# Audiometric Tests without Booths

**DOI:** 10.3390/ijerph18063073

**Published:** 2021-03-17

**Authors:** Alberto Behar

**Affiliations:** Department of Psychology, Ryerson University, Toronto, ON M5B 2K3, Canada; albehar31@gmail.com

**Keywords:** background noise, audiometric booth, audiometric tests, false positive results

## Abstract

Audiometric booths are used to reduce background noise levels at testing locations to below values specified in the standards. As such, they are considered inherent parts of the audiometric testing equipment. This paper presents the results from a literature search of solutions that could ensure that background noise levels are acceptable outside booths. The technology used is especially valuable for survey tests and for locations where booths are unavailable or cannot be used for different reasons. However, its use is recommendable for only screening hearing tests but not for clinical or research applications.

## 1. Introduction

Audiometric booths constitute part of the basic equipment conventionally thought to be required to perform hearing tests. The booth is to reduce background noise and, as a consequence, the number of false positive results. The booth also diminishes the opportunity for distraction of the examinee.

A low background noise is of such an importance as to be included in the standards for audiometric tests. As an example, the latest Canadian Standard Association (CSA) Standard for audiometric tests [1,2], reproduced here as Table 1, shows the maximum permissible ambient sound levels at the different octave bands ranging from 125 to 8000 Hz. Both types of earphones, included in the table, are effective at reducing background noise. However, insert earphones are slightly more effective than supra-aural, especially at low frequencies. As an example, at 125 Hz, the allowable background noise for supra-aural earphones is 49 dB, whereas inserts allow up to 78 dB at the same frequency

The values in Table 1 are the maximum allowed, meaning that they should not be exceeded at any time. If such a situation occurs, the test has to be interrupted and resumed once the noise levels have been reduced below the limit. This is a situation that can occur in locations close to a corridor with audible traffic noise or with windows facing a street. This is also a frequent problem with mobile audiometric facilities located in parking lots.

There are several problems associated with the use of audiometric booths. Probably the most important is their cost. In many cases, it exceeds the cost of the audiometer! Single-wall booths can be installed with substantial savings, but they are only suitable in a relatively quiet place or when they are used for routine survey tests. However, in the cases of clinical examinations, for research purposes, or if the testing location is close to a shop floor, a double-wall device may be needed, resulting in higher costs. 

Another issue is the floor loading caused by the weight of the booth, as well as the space that it occupies. In general, more expensive booths require more space, and they tend to be heavier. If there is a need to also isolate the audiometer operator, then a double room is needed, and even more space has to be allocated.

Another big issue is the lack of compliance with the permissible ambient noise levels. A study by Frank and Williams [3] measured sound levels in 136 audiometric rooms in various audiological facilities. They found that only 50% of the booths complied with the requirements for testing between 250 to 8000 Hz. 

It is also important to consider the psychological factors associated with testing in audiometric booths that may result in negative experiences for the participants and disrupt the testing procedures. Claustrophobia is a situational phobia triggered by an irrational fear of tight or crowded spaces [4]. It can be triggered by situations like being locked in a windowless room, being stuck in a crowded elevator, or in a cavern. Some people tend to feel claustrophobic and highly uncomfortable in an audiometric booth, especially when the examiner dims the lighting to help the participant concentrate. 

The most serious issue when dealing with booths is how to ensure that the noise inside is within the recommended limits at all times. As pointed out, this requirement becomes difficult to comply with in locations with variable noise levels—that is, a situation found at most testing locations that are in or close to an industrial site or parking lot. Production and traffic noises are paramount in those locations. It makes complying with the above-mentioned standard [1] (and, as a matter of fact, all similar standards) problematic. In such situations is when the requirement for the background noise to be observed continuously during testing, as well as be tested at least once a year, becomes so important, up to the point of recommending the use of devices that monitor continuously and alert the operator when the levels exceed the noise limits.

## 2. Discussion

### 2.1. Tests Without Booths?

Is the use of a booth essential? The title of Table 1 is “Maximum permissible ambient noise levels…”, without specifically mentioning “inside audiometric booths”. In other words, the issue is the value of the ambient noise level outside the earphone and not the way it is obtained. This point underpins the different booth-less solutions for performing audiometric tests that have been attempted in the last 50 or so years, some of which are reviewed here. The feasibility of these alternative solutions have been assessed by comparing the booth-less assessment results to those obtained from tests performed inside a booth. 

One of the first attempts to conduct audiometry outside of a booth was done in 1967 by R.R.A. Coles [5]. His approach was to increase the attenuation of the almost universally used earphone: the Telephonics TDH-39 receiver equipped with a MX-4I/AR cushion. He attempted to enclose the earphone using a proprietary muff. The increase of the attenuation compared to the regular TDH-39 was of some 10 dB across the frequency range 250 Hz–8 KHz. The newly designed headphone was intended for the audiometric screening of recruits for the British Royal Navy and Royal Marines, in view of the advantage over audiometric booths in terms of cost, space, and floor-loading. The author recognized that there were problems with the calibrations. 

Although this first attempt did not result in a viable solution, it opened the discussion on the feasibility of booth-less options for audiometric testing. Following, there have been several booth-less solutions using different techniques to reduce background noise that show promise. It is worth noting, however, that most authors recognize that their solutions apply for screening tests, where the objective is the detection of hearing levels larger than 0 dB Hearing TYhreshold Level (HTL). They recommend the booth-less solutions for rural areas, health clinics, schools, or nursing homes, where environmental noises are low and where sound booths are not readily available. Brief descriptions of some these, as well as the techniques and results obtained, are provided in the sections that follow.

### 2.2. Tests in a Quiet Environment or with Enhanced Sound Attenuation Headphones

Rural environments or small villages tend to have fairly low ambient noise levels. This author remembers auditing audiometric tests in a Bauxite processing facility in Guinea where tests were performed in an office, with no audible environmental noise (not measured) [6]. The results of the audiometric tests showed very little or no hearing loss. Obviously, this was an ideal location for performing tests without a booth. However, this is not the only circumstance in which testing without a booth may be appropriate. Although testing locations in health (or similar) clinics may not comply with the standard requirements for ambient noise levels, they may still be adequate for routine screening procedures, especially when the objective is to detect hearing loss in excess of 25 dB. As an example, Wong et al. [7] performed audiometric tests on 885 transport workers in both nonsoundproofed and soundproofed environments to compare the level of agreement between the results. Those who were found to have a hearing loss of more than 25 dB in either ear at any of the frequencies tested were requested to attend a conventional diagnostic, pure-tone audiometric test in a soundproofed booth. The authors concluded that the results of the tests conducted in nonsoundproofed environments in the field were comparable to those obtained in a soundproofed environment. 

On the other hand, where compliance with the standard requirements is necessary in nonclinical or research settings, enhanced sound attenuating headphones may be sufficient for obtaining valid hearing thresholds. The Wireless Automated Hearing Test System (WAHTS) is an innovative mobile, wireless system that provides the opportunity to measure hearing loss in the field without the use of booths. The WAHTS system includes a pair of specially designed earphones, where the speaker is enclosed in a large, rigid shell, lined with thick polyurethane foam. Using a within-subjects design, Meinke et al. [8] tested the hearing of industrial workers to assess the test–retest reliability of hearing thresholds obtained with WAHTS at a booth-less worksite when compared to those obtained in a standard automated mobile van equipped with sound booths. For the booth-less WAHTS tests, the subjects were tested in conference rooms at different locations within the worksite building. According to the authors, this initial study demonstrated that the attenuation afforded by the headphones of the WAHTS was sufficient to obtain valid hearing thresholds in diverse workplace test locations without the use of sound-attenuating enclosures.

Recently, researchers have furthered remote testing by developing hearing assessment software that may be integrated with smartphone technology. This could be extremely useful in underserved community healthcare clinics. Sandström et al. [9] performed a study to validate a calibrated smartphone-based hearing test when performed in both sound booth environments and in primary healthcare clinics. To do so, 64 subjects were tested in a sound booth and 30 were tested in clinics without a booth. According to the authors, this study provided the first evidence of accurate air conduction hearing thresholds determined by an inexpensive smartphone (Android OS) and off-the-shelf supra-aural headphones calibrated according to the international standards. They concluded that accurate air conduction audiometry can be performed without a sound booth using a smartphone, especially in cases where the measured thresholds exceed 15 dB HTL. 

### 2.3. Combination of Insert Earphones and Over-the-Ear Earmuffs

The CSA standard [1] allows for significantly higher ambient noise levels at all frequencies when insert headphones are used. Additionally, it is well-known that well-fitted, over-the-ear earmuffs significantly reduce the ambient noise. Using both at the same time, the ambient noise is reduced twofold: by the insert earphone and by the earmuff. 

In a study, Buckey et al. [10] examined the possibility of obtaining reliable threshold measurements without a sound booth by using a passive hearing protector (David Clark Model 10 A earmuff) combined with in-ear 1/3-octave band noise measurements. An in-ear probe (used for DPOAE tests) containing a microphone and the earphone were used under the hearing protector for both the in-ear noise measurements and threshold audiometry. The study included three cohorts: two in Tanzania (one of adults and another pediatric) and one in the USA. The Tanzanian adult cohort included 624 subjects (400 females and 224 males) with an average age of 39 years. The Tanzanian pediatric cohort included 197 subjects (55% female) with an average age of 10 years. The United States cohort had 100 normal hearing individuals (64% females, average age of 39 years). No information regarding the testing environment was provided in the paper. According to the authors, the results showed that an adequate environment for threshold audiometry can be created without a sound booth by testing the in-ear sound levels.

Another option for the combination of insert earphones and over-the-ear earmuffs is the recently developed KUDUwave (eMoyo Technology) [11], a computer-based audiometer. It uses insert earphones and large circumaural, specially designed muffs with increased noise attenuation. Each cup is fitted with a microphone that monitors the noise levels inside the muff, allowing the operator to interrupt the test if the levels exceed the pre-set limits. The audiometer is intended for screening and clinical testing outside a sound booth.

Swanepoel et al. [12] used a KUDUwave audiometer to compare the hearing of 149 school children from two schools (average age 6.9 years), as tested in an audiometric room and in a natural school environment. The tests were first conducted in rooms provided by the schools (a classroom, administrative room, or media room). Then, the same evaluation was done within a few days of the initial trial in an audiometric booth located in an audiological clinic. The average differences between the measurements done in a natural environment and in an audiometric booth were between 2.02 dB and 0.5 dB, with standard deviations between 2.5 dB and 4.7 dB across the frequencies and ears. The authors concluded that the results confirmed statistically and clinically equivalent hearing thresholds for children tested in a school environment compared to a sound-treated booth. Again, there was no data regarding the sound levels in the school rooms during the tests.

In another study, Swanepoel et al. [13] investigated the feasibility of the KUDUwave™ Plus audiometer for clinical testing outside a sound booth. The study comprised two parts. The first was to study the attenuation characteristics of the muff using the ambient noise-monitoring feature of the audiometer. For this purpose, HTLs of 15 normal hearing subjects were measured using a regular audiometer in a sound booth. Then, the test was repeated with audiometric signals transmitted through external loudspeakers three times, while the ears of the subjects were protected as follows: (a) by using TDH39 headphones, (b) inserted earphones, and (c) a combination of inserted earphones and the KUDUwave earmuffs. The attenuation of the earmuffs was calculated as the difference between the ambient SL and the SL, measured by the microphones enclosed in the earmuffs. 

The second study consisted of testing the accuracy and reliability of audiometric tests outside a sound booth using the KUDUwave audiometer. For that purpose, 23 normal hearing subjects were tested—first, with a conventional audiometer in a double-wall audiometric booth and, then, using the KUDUwave audiometer in a regular office environment (ambient noise levels around 46 dB across the frequencies 250–8000 Hz). The authors concluded that accurate and reliable testing in a natural environment (outside of a booth) using automated testing is comparable to that of manual audiometry conducted within a sound booth when using sufficient earphone attenuation in combination with the real-time monitoring of ambient noise.

Maclennan-Smith et al. [14] used the KUDUwave™ Plus to compare the pure-tone conducted thresholds (250–8000 Hz) measured in retirement facilities with the thresholds measured in a sound-treated booth. One hundred and forty-seven adults (average ages 76 ± 5.7 years) were evaluated. The pure-tone averages were ≥25 dB in 59%, mildly elevated (>40 dB) in 23%, and moderately elevated (>55 dB) in 6% of the ears. The measured thresholds (*n* = 2259) corresponded within 0–5 dB in 95% of all comparisons between the two test environments. The average threshold differences (−0.6 to 1.1) and standard deviations (3.3 to 5.9) were within the typical test–retest reliability limits. The thresholds recorded showed no statistically significant differences (paired samples *t*-test: *p* > 0.01), except at 8000 Hz in the left ear. The authors concluded that valid diagnostic pure-tone audiometry can be performed in a natural environment with the technology they used. This is significant, as it offers the possibility of access to diagnostic audiometry in communities where sound-treated booths are nonexistent.

### 2.4. Active Noise Reduction (ANR) Earmuffs

The well-known ANR electronic technology relies on the principle of destructive interference to cancel noise. For the purpose of noise cancelation, a control microphone is located under the muff’s cup. It picks up the noise that has penetrated the muff. The signal’s phase is then rotated 180^0^, amplified, and fed to a speaker also located under the muff. In theory, the noise that is fed back should completely destroy the original noise. Due to several practical limitations, the result is a significant reduction of the low-frequency noise level, below 1 KHz, that has penetrated the earmuff. However, there is little effect on the higher frequencies. Nonetheless, the control of these low frequencies is still quite helpful, since they are known to cause forward masking (masking of a signal of frequency higher than the interfering one).

Clark et al. [15] studied the feasibility of using insert earphones in combination with an ANR earmuff (Bose QuietComfort15) for audiometric evaluations in out-of-booth audiometric assessments (e.g., tele-audiologic practice in rural/community settings). For that purpose, the authors calculated the maximum acceptable environmental sound level by adding the attenuations of the muff and the earphones to the levels specified in the ANSI 1999 Standard [16]. The results of their study suggested that ANR muffs (specifically, the Bose QuietComfort 15) can sufficiently reduce the ambient noise in a typical patient consultation room to well below the ANSI maximum permissible sound levels for audiometric air conduction testing.

Saliba et al. [17] conducted a study with two main objectives: (a) compare the accuracy of previously validated mobile-based hearing tests in determining the pure-tone thresholds and screening for hearing loss and (b) determine the accuracy of mobile audiometry in noisy environments through noise reduction strategies. For the first purpose, 33 subjects were subjected to four different hearing tests as follows: (1) standard tests with insert audiometric earphones in a double-walled sound booth, (2) self-administered tests in a sound booth using the EarTrumpet (Apple store) app, (3) self-administered tests in a sound booth using the SHOEBOX (Apple store) app, and (4) self-administered tests in a sound booth using any of the above apps inside a 50-dBA ambient noise environment. On top of the insert earphones, the subjects donned ANC muffs Noisebuster PA 4000 (Pro Tech Technologies), and the tests were alternated with and without the active noise cancellation on. The authors concluded that mobile audiometric applications can correctly measure pure- tone thresholds and screen for moderate hearing loss. The addition of noise reduction strategies provides a portable and effective solution for hearing assessments in various settings inside quiet clinics. However, when abnormal audiometric screening is to be identified, as per the authors, formal audiometry is still required.

Bromwich et al. [18] oriented their study around the problem posed by low-frequency background noise, identified as presenting a specific limitation for screening audiometry. This is a problem even in circumstances where insert earphones and portable single-walled sound booths are used. Low-frequency noises and vibrations are significant factors often found in industrial and institutional settings. The study comprised two parts. For the first part, the authors measured the sound attenuation of a set of Bose Aviation X circum-aural ANR headphones (Bose, Framingham, MA, USA). This was done using a head and torso simulator in a semi-anechoic chamber. The attenuation was measured with the ANR on and off and, also, with and without the inserted earphones used for the audiometric tests. The second part of the study consisted of audiometric tests (250–4000 Hz) performed on 10 subjects in a sound booth with background noises of 0, 30, and 40 dBA. The authors suggested that the results from their study confirmed the feasibility of using a combination of inserted earphones and ANR muffs for performing audiometric tests even in moderately noisy conditions and, especially, when the noise is a low-frequency content.

## 3. Conclusions

Several solutions, as described in this paper, have been proposed that allow for audiometric testing in the absence of audiometric booths. They tend to incorporate a quiet test environment and the use of a combination of earmuffs and insert earphones, as well as the use of ANR. Research studies using some of these solutions suggest that they may be appropriate for low sound-level environments and for routine surveillance audiometric tests. Unfortunately, many of these studies provided scant details regarding the noise levels in their testing environments, something important for the validation of their results. What the studies had in common was an agreement about the need of a booth when the tests are for clinical or research purposes. Additionally, they emphasized the need for permanently monitoring the ambient noise levels, whether or not an audiometric booth is used. With that said, there are circumstances, as outlined in this paper, where booth-less solutions may be appropriate and can be used to increase the access to hearing assessments in remote settings.

## Figures and Tables

**Table 1 ijerph-18-03073-t001:** Maximum permissible ambient noise levels for audiometric testing with supra-aural and insert earphones used for testing in the frequency range 500 to 8000 Hz.

	Maximum Level (dBSPL)	
Octave Band (Hz) *	Supra-Aural Earphones	Insert Earphones
125 ^†^	49	78
250 ^†^	35	64
500	21	50
1000	26	47
2000	34	49
4000	37	50
8000	37	56

* The maximum noise levels by octave band specified in this table are from ANSI/ASA S3.1. ^†^ Audiometric testing at pure-tone frequencies below 500 Hz, if conducted, requires lower maximum ambient noise levels at octave bands of 125 Hz and 250 Hz than those specified in this table. See ANSI/ASA S3.1.

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
