# Peer review of "Audiometric Tests without Booths"

_ijerph, 2021, doi:10.3390/ijerph18063073_

Round 1
Reviewer 1 Report
I would like to congratulate the author for this study.
However, several points as indicated below need to be addressed by authors to improve the quality of the article:
- In the abstract you mentioned: However, its use is not recommendable for clinical or research applications. If it is not to be applied in the clinical practice or in research, in what situations can it be applied? For example, in screening hearing tests? This question should be clarified for the reader to understand the importance and application of this study.
- I would propose an improvement of the abstract, making it more developed and with a conclusion that highlights the importance of this manuscript.
- In Table 1, the values of 49 and 78dB with supra-aural and insert earphones, respectively, are relative to what frequency?125Hz?
- In the title of Table 1 you mentioned: Maximum permissible ambient noise levels for audiometric testing with supra-aural and insert earphones used for testing in the frequency range 500 to 8000 Hz. However, in the table we can check relative data at frequencies below 500Hz. Can you clarify this situation, please?
- This manuscript is based on a review of the literature on the subject under analysis. It would be important to include material and methods where it could place the databases consulted, the search engines used, the number of articles used for this review, the time/temporal period of those articles, exclusion and inclusion criteria, ….
- It makes reference to different studies that use Tests in a silent environment.In this test condition, which frequencies are tested?Even in situations where the ambient noise is reduced, isn't it risky to test the 250 and 500Hz? Even for hearing screening.
- The Wireless Automated Hearing Test System (WAHTS) seems to be very promising.In addition to the study of Meinke, D. K. and collaborators, what other studies have used this system? This innovation should be supported by different studies.
- During the manuscript you mentioned: no data regarding the sound levels in the school rooms during the tests. However, you mentioned in the conclusion: Unfortunately, many of these studies provide scant details regarding the noise levels in their testing environments, something important for the validation of their results. Shouldn't it be considered a study limit?
- The numeration of different parts in this manuscript can be reviewed. For example, discussion was the second point and the next point is the conclusion with number five. Will it not be the third?
Author Response
Please see the attachment。

Reviewer 2 Report
In initial review of the manuscript I found it to be generally well-written, largely acceptable as-is. I don't think it is breaking any new ground; it appears to be a meta-analysis of existing research in this area. I read the paper straight through twice with very few notes or comments.
The limited comments I did make included the following:
1. "CSA" is mentioned in the second paragraph of the introduction. I assumed this to mean "Canadian Standards Association" - and it is indicated in the citation, but it should be written out in its first mention in the paper.
2. table 1: should the first entry under "Octave Band (Hz)" be "125" rather than "4"?
3. Page 2, end of second paragraph: "...when the examiner switches off the lighting..." I am unaware of anyone taking a hearing test in the dark. Perhaps "...dims the lighting..." would be more accurate?
Otherwise I have no comments aside from that this is a well-written paper. Good review of existing literature.
Reviewer 3 Report
Thank you for sharing your work. The paper is well written to support the proposition that in most cases audiometric tests can be performed without expensive booths setup.
I have revised the manuscript and my comments are as follows:
1. Section 2.2, last paragraph: What is meant by " ... primary health care clinics" ?
2. Section 2.2, last paragraph, last line: Does the author mean to say "" .... measured threshold doesn't exceed 15 dB HTL"
3. Section 2.3: It seems reference [10] & [11] may not be reliable as data regarding ambient noise is missing. Please comment on that.
